# Cardiac Cx43 and ECM Responses to Altered Thyroid Status Are Blunted in Spontaneously Hypertensive versus Normotensive Rats

**DOI:** 10.3390/ijms20153758

**Published:** 2019-08-01

**Authors:** Matus Sykora, Barbara Szeiffova Bacova, Tamara Egan Benova, Miroslav Barancik, Jitka Zurmanova, Hana Rauchova, Peter Weismann, Stanislav Pavelka, Lin Hai Kurahara, Jan Slezak, Tomas Soukup, Narcis Tribulova

**Affiliations:** 1Institute for Heart Research, Centre of Experimental Medicine, Slovak Academy of Sciences, 841 04 Bratislava, Slovak Republic; 2Department of Physiology, Faculty of Science, Charles University, 128 00 Prague, Czech Republic; 3Institute of Physiology, v.v.i., Academy of Sciences of the Czech Republic, 142 20 Prague, Czech Republic; 4Institute of Anatomy, Faculty of Medicine, Comenius University in Bratislava, 811 08 Bratislava, Slovak Republic; 5Institute of Molecular Genetics, v.v.i., Academy of Sciences of the Czech Republic, 142 20 Prague, Czech Republic; 6Department of Cardiovascular Physiology, Faculty of Medicine, Kagawa University, Kagawa 761 0793, Japan

**Keywords:** thyroid hormones, spontaneously hypertensive rats, heart, connexin-43, extracellular matrix

## Abstract

Heart function and its susceptibility to arrhythmias are modulated by thyroid hormones (THs) but the responsiveness of hypertensive individuals to thyroid dysfunction is elusive. We aimed to explore the effect of altered thyroid status on crucial factors affecting synchronized heart function, i.e., connexin-43 (Cx43) and extracellular matrix proteins (ECM), in spontaneously hypertensive rats (SHRs) compared to normotensive Wistar Kyoto rats (WKRs). Basal levels of circulating THs were similar in both strains. Hyperthyroid state (HT) was induced by injection of T3 (0.15 mg/kg b.w. for eight weeks) and hypothyroid state (HY) by the administration of methimazol (0.05% for eight weeks). The possible benefit of omega-3 polyunsaturated fatty acids (Omacor, 200 mg/kg for eight weeks) intake was examined as well. Reduced levels of Cx43 in SHRs were unaffected by alterations in THs, unlike WKRs, in which levels of Cx43 and its phosphorylated form at serine368 were decreased in the HT state and increased in the HY state. This specific Cx43 phosphorylation, attributed to enhanced protein kinase C-epsilon signaling, was also increased in HY SHRs. Altered thyroid status did not show significant differences in markers of ECM or collagen deposition in SHRs. WKRs exhibited a decrease in levels of profibrotic transforming growth factor β1 and SMAD2/3 in HT and an increase in HY, along with enhanced interstitial collagen. Short-term intake of omega-3 polyunsaturated fatty acids did not affect any targeted proteins significantly. Key findings suggest that myocardial Cx43 and ECM responses to altered thyroid status are blunted in SHRs compared to WKRs. However, enhanced phosphorylation of Cx43 at serine368 in hypothyroid SHRs might be associated with preservation of intercellular coupling and alleviation of the propensity of the heart to malignant arrhythmias.

## 1. Introduction

Heart function is profoundly regulated by thyroid hormones (THs). The biologically active form of TH, 3,5,3′-triiodo-l-thyronine (T3), modulates cardiac electrophysiology and calcium handling via both genomic and non-genomic mechanisms [1,2,3,4,5]. Changes in circulating THs influence the basal activity of cardiomyocytes whereby the excess or deficiency of THs, resulting from thyroid dysfunction, is harmful to the heart [1,6,7,8]. Overt thyroid disease is prevalent at older ages while its subclinical form is prevalent in young adults [4,9]. Cardiac disease itself may alter TH status (notably, low T3 syndrome) and contribute to heart failure [5].

TH may affect the function of the heart [10], and its susceptibility to arrhythmias [2,4] as well as facilitate sinus rhythm restoration [11]. Unlike hypothyroidism, hyperthyroidism in humans promotes atrial fibrillation (AF) [12]. In animal models [13], the hyperthyroid state increased and hypothyroid state decreased the propensity of the heart to ventricular fibrillation (VF). Hyperthyroid rats were also more prone to developing AF [14]. The propensity to these arrhythmias was related to the myocardial protein level of connexin-43 (Cx43), which is downregulated because of excess THs and upregulated in TH deficiency [14,15,16,17]. Defects in protein abundance and distribution of Cx43 have been known to impair Cx43 channel-mediated intercellular electrical coupling, which is crucial in arrhythmogenesis [18,19].

With regard to arrhythmias, an excess of THs is known to induce cardiac hypertrophy but not fibrosis [15,17,20,21]. Fibrosis resulting from hypertension, post-infarction, or other pathologies deteriorates electrical coupling and promotes arrhythmias and contractile dysfunction [22,23,24,25,26]. Thus, it is of great importance to understand the molecular mechanisms of extracellular matrix (ECM) regulation by THs in the heart during physiological as well as pathophysiological conditions. Unlike hypertension-related cardiac hypertrophy, hyperthyroidism-related hypertrophy is not associated with enhanced expression of β-myosin heavy chain (*MYH7*) and suppression of sarcoplasmic Ca2+ ATPase *(SERCA2*) [3,17,27], which may contribute to heart failure and contractile dysfunction.

Considering these differences, we hypothesized that hypertensive rats may differ in responsiveness to altered circulating THs compared to normotensive rats. In the context of the susceptibility of the heart to arrhythmias, we focused on exploring myocardial Cx43 and proteins involved in ECM turnover, such as matrix metalloproteinase-2 (MMP-2), transforming growth factor β1 (TGF-β1), and TGF transcription factor (SMAD2/3) in spontaneously hypertensive rats (SHRs) and Wistar-Kyoto rats (WKRs) in conditions of TH excess and deficiency. We also tested the effect of omega-3 polyunsaturated fatty acids intake, using Omacor, which has been recently assessed for its cardiac antiarrhythmic effects [28]. 

## 2. Results

### 2.1. Characteristics of Experimental Rats

The biometric parameters and serum TH levels of experimental rats are summarized in Table 1. There were no differences in basal TH levels between WKRs and SHRs. However, the thyroid gland weight (TGW) was lower in euthyroid SHRs than in euthyroid WKRs, and increased in both strains due to the treatment with methimazole. Hyperthyroid status did not affect this parameter significantly. Decrease or increase in circulating total T3 and total thyroxine (T4) due to the treatment with methimazole or THs confirmed hypothyroid and hyperthyroid status in both strains. Omacor intake decreased T4 in hyperthyroid WKRs, while the reduction was not significant in SHRs likewise the decline of T3 in both strains.

There were no significant changes in blood glucose and body weight among the experimental groups. Omacor intake reduced the body weight of euthyroid and hyperthyroid WKRs. Hyperthyroidism was accompanied by an increase in heart and left ventricular weights in WKRs and SHRs, whereas hypothyroidism decreased both parameters regardless of the rat strain. Determination of myocardial thiobarbituric acid reactive substances (TBARSs) revealed that this parameter was significantly increased only in hyperthyroid WKRs, while intake of Omacor had a decreasing effect.

The serum lipid profile of experimental rats is presented in Table 2. There were no significant changes in triacylglycerol, high-density lipoprotein cholesterol (HDLchol), and low-density lipoprotein cholesterol (LDLchol) levels, and the ratio of total cholesterol and HDLchol between WKRs and SHRs. Hypothyroid status significantly increased both HDLchol and LDLchol in WKRs as well as in SHRs. Omacor intake did not affect the lipid profile in any experimental group.

### 2.2. Myocardial MYH7, SERCA2, and PPARγ mRNA Transcripts 

Levels of myocardial *MYH7*, *SERCA2,* and peroxisome proliferator-activated receptors gamma (*PPAR**γ***) gene transcripts in experimental rats are summarized in Table 3. *MYH7* transcript corresponding to β-MyHC was significantly increased in SHR, compared to WKR. Hyperthyroid status decreased whereas hypothyroid status increased this parameter in both rat strains significantly. There was no significant difference in *SERCA2* transcript expression between rat strains. Hypothyroid status decreased this parameter in WKR as well as in SHR, whereas hyperthyroid status showed a tendency to increase *SERCA2* mRNA expression. There was no difference in *PPAR**γ*** mRNA expression between euthyroid WKR and SHR. However, this transcript was significantly upregulated in hyperthyroid WKR, and to a lesser extent in hyperthyroid SHR, compared to the euthyroid rats. Omacor intake did not affect significantly the expression of any examined gene transcripts.

### 2.3. Myocardial Cx43 mRNA Transcript and Protein Levels

Real-time PCR analysis revealed that altered thyroid status did not affect *CXA1* mRNA in WKRs and SHRs (Figure 1A), although there was an apparent strain-related difference. The expression of *CXA1* mRNA was significantly higher in SHRs than in WKRs. Levels of total Cx43 protein (Figure 1B,C), as well as its phosphorylated form at serine368 were significantly lower (Figure 2A,B), whereas levels of the non-phosphorylated form were higher (Figure 2C,D), in SHRs than in WKRs. Altered thyroid status did not affect Cx43 protein levels in SHRs, unlike in WKRs. Hyperthyroid status decreased, while hypothyroid status increased, total Cx43 protein levels (Figure 1B,C). As Cx43 is active in the phosphorylated form, its obvious phosphorylation at serine368 residues was increased due to hypothyroid status in both strains when compared to euthyroid rats (Figure 2A,B). The hyperthyroid state did not suppress Cx43 phosphorylation in SHR hearts unlike in WKR hearts (Figure 2A,B). Hypothyroid status increased phosphorylation of Cx43 at serine368 (Figure 2A,B) as well as non-phosphorylated Cx43 in SHRs (Figure 2C,D). Myocardial in situ immunolabeling showed the conventional distribution of abundant Cx43-positive fluorescence signal at the intercalated disc-related gap junctions in all experimental groups (Figure 3 WKRs, Figure 4 SHRs). In addition, Cx43 labeling was only sporadically observed on the lateral sides of the cardiomyocytes in euthyroid WKRs, while it was more frequently observed in euthyroid SHRs as well as in hypothyroid WKRs and SHRs. Intake of Omacor did not significantly affect *CXA1* transcript, protein levels and distribution in any experimental group.

### 2.4. Myocardial PKC-ε mRNA Transcript and Protein Levels

Protein kinase C-epsilon (PKC-ε) phosphorylates Cx43 protein at serine368 on the carboxyl-terminus of Cx43. The myocardial level of this protein kinase was significantly lower, whereas that of its transcript was higher in SHRs, than in WKRs (Figure 5A–C). Both *PKC-ε* mRNA expression and protein levels were increased in hypothyroid WKRs and to a lesser extent in hypothyroid SHRs. However, PKC-ε protein abundance was suppressed due to hyperthyroid status, which was significant only in WKRs when compared to euthyroid rats. Omacor intake did not show significant changes *in PKC-ε* mRNA expression and protein level in either group. 

### 2.5. Myocardial TGF-β1, SMAD2/3, and MMP-2 Protein Levels

Basal TGF-β1 protein abundance was higher in WKRs than in SHRs (Figure 6A,B) and a significant decrease was observed in hyperthyroid WKR while there was no change in hypothyroid WKRs. Unlike in the WKR heart, altered thyroid status did not affect TGF-β1 protein abundance in the SHR heart. There were no significant differences in protein levels of SMAD2/3 between WKRs and SHRs (Figure 6C,D). Hyperthyroid state decreased the SMAD2/3 protein level whereas the hypothyroid state led to a significant increase in SMAD2/3 protein levels in WKRs. In SHRs, an increase in SMAD2/3 protein abundance was observed only in the hypothyroid state. The protein abundance of 72 kDa pro-MMP-2 was significantly higher in euthyroid SHRs than WKRs and alterations in thyroid status did not affect MMP-2 protein level in either WKR or SHR hearts (Figure 7A–C). Omacor intake did not significantly affect MMP-2 protein abundance in any experimental group.

### 2.6. Myocardial MMP-2 Activity 

There were no strain-related differences in basal activity of either forms of MMP-2 (Figure 8A–C). Hyperthyroid status decreased the levels of 72 kDa MMP-2 along with an increase in 63 kDa MMP-2 activity in WKRs. However, hypothyroid status decreased the activity of 72 kDa MMP-2 and partially also of 63 kDa MMP-2 in WKRs. In contrast, altered thyroid status did not significantly affect MMP-2 activity in SHRs compared to euthyroid rats. Omacor intake suppressed the levels of 63 kDa MMP-2 in hyperthyroid WKRs and euthyroid SHRs.

### 2.7. Myocardial Collagen Deposition 

As shown in Figure 9, the red staining attributed to collagen deposition was scarcely present in hyperthyroid but was enhanced in the hypothyroid WKR heart left ventricle. Quantitative image analysis (Figure 9) further confirmed that the collagen content was significantly enhanced in hypothyroid but unchanged in hyperthyroid rats, compared to euthyroid WKRs. Furthermore, Omacor intake suppressed collagen production in hypothyroid WKRs.

In contrast to the WKR heart, collagen-specific staining was increased in the SHR heart regardless of thyroid status as demonstrated in Figure 10. As shown in this panel, there were no significant differences between hyperthyroid and hypothyroid SHR hearts when compared to euthyroid rats, irrespective of Omacor intake.

## 3. Discussion

In agreement with our hypothesis, we have demonstrated that despite similar basal levels of circulating THs, there were differences between hypertensive and normotensive rat hearts in response to altered thyroid status. In general, there were significant changes in myocardial Cx43 levels and markers of ECM in 12-month-old hyperthyroid as well as hypothyroid WKRs; in contrast, no apparent change was observed in the corresponding SHRs. This could be either explained by abnormalities in the TH negative feedback regulation of thyroid stimulating hormone secretion in SHRs [29] and/or by counteracting molecular signaling associated with essential hypertension. Hence, these differences warrant further investigation. Nevertheless, the key findings point out that thyroid status may profoundly affect the heart function and its susceptibility to arrhythmias in WKRs but to a lesser extent in SHRs.

Questions arise regarding whether genomic or non-genomic (or both) signaling of thyroid hormones is blunted in SHR. The transcript of Cx43 was not altered by thyroid status regardless of the strain. However, Cx43 protein levels were increased in hypo- and decreased in hyperthyroid WKR, while not in SHR. Likewise functional phosphorylated forms of Cx43 were not suppressed in hyperthyroid SHR unlike in WKR. These facts suggest that responsiveness of SHR heart to altered thyroid status is most likely hampered post-translationally. To further understand the relationship, it will be necessary to include wider spectrum of analysis, namely phosphorylation of Cx43 by various protein kinases. Phosphorylation of Cx43 is known to affect profoundly Cx43 turnover, which is fast (half-life 1.5 h) in the heart [18]. Another factor implicated in the hampered response of SHR to alterations of TH could be the difference in fatty acids composition, namely a lower content of omega-3 polyunsaturated fatty acids, as reported in cell membranes of SHR versus Wistar rats [28]. Such alterations may modulate membrane function and signal transduction.

An increase in heart and left ventricular weights due to excess THs, and decrease due to TH deficiency was observed in WKRs as well as SHRs. Both strains also exhibited enhanced *SERCA2* expression and suppressed *MYH7* expression in hyperthyroid as opposed to hypothyroid status. Our findings using WKRs correspond with those reported in other studies using Wistar rats [13,17,30], while no reports on SHRs have yet been published. It appears that TH deficiency may accelerate heart failure in SHRs due to the decrease in *SERCA2* and increase in *MYH7* expression. On the other hand, the upregulation of *SERCA2* and downregulation of *MYH7* due to hyperthyroidism may hamper heart contractile dysfunction in SHRs. Moreover, upregulation of *SERCA2* in SHRs may attenuate Ca^2+^ overload, which is known to promote triggered activity [31] and electrical uncoupling at the Cx43 channels [32]. 

As already noted, altered thyroid status did not affect *CXA1* mRNA transcripts in WKRs or SHRs, with SHRs exhibiting higher basal levels of *CXA1* mRNA than WKRs or Wistar rats, as shown in previous study [33]. Nevertheless, total Cx43 protein levels were enhanced in hypothyroid but suppressed in hyperthyroid WKRs, as similarly reported in Wistar rats [13,17]. In accordance with the findings of previous studies, levels of Cx43 protein were lower in SHR hearts than in normotensive rat hearts [27,33]. Unlike WKRs, altered thyroid status did not affect Cx43 protein abundance in SHRs despite comparable values of circulating T3 and T4 in both strains. This most likely suggests distinct molecular signaling involved in Cx43 turnover in essential hypertension that is unaffected by TH excess or deficiency.

In parallel with Cx43 protein, the protein levels of its functional phosphorylated form at serine368 residue (P-Cx43,) that stabilizes conduction [34] were decreased in hyperthyroid, and increased in hypothyroid, WKRs. VF-prone SHR hearts exhibited lower basal levels of P-Cx43 and higher levels of non-phosphorylated Cx43 (N-Cx43) than those observed in hearts of normotensive rats, which was consistent with previous studies [27,33,35]. Interestingly, SHR hearts did not respond to excess THs while TH deficiency increased both P-Cx43 and N-Cx43 protein levels. These changes might be beneficial in protection from arrhythmias. An increase in phosphorylated Cx43 at serine368 by PKC-ε [36] can prevent the abnormal distribution of Cx43 and stabilize Cx43 channel conductivity [34]. Indeed, higher protein abundance of Cx43 and its phosphorylated form at serine368 were associated with reduced incidence of VF [37,38]. This may explain, at least in part, the lower susceptibility of hypothyroid and higher susceptibility of hyperthyroid Wistar rats to malignant arrhythmias as shown previously [13].

It should be emphasized that Cx43 channel function is regulated by its phosphorylation. One of the protein kinases that phosphorylates Cx43 is PKC-ε [39]. Interestingly, *PKC-ε* transcript was higher while protein levels lower in SHR versus WKR perhaps reflecting some disorder in translation. However, PKC-ε protein abundance was enhanced due to TH deficiency in WKRs as well as in SHRs. An increase in PKC-ε in hypothyroid rats and decrease in PKC-ε in hyperthyroid rats corresponded with alterations in P-Cx43. The antiarrhythmic PKC-ε-Cx43 pathway that was hampered by treatment with THs has also been shown in diabetic rats [15]. The protein abundance of PKC-ε was lower in SHRs than in WKRs; however, the protein abundance was enhanced in hypothyroid rats, thus indicating the potential benefit. Taken together, our findings suggest that besides other factors, the level of Cx43 phosphorylation at serine368 most likely by PKC-ε might be implicated in anti- (in HY status) and pro-arrhythmic (in HT status) signaling. This hypothesis that we already postulated previously [14,15,16,37] was strongly supported by the results of current study.

With regard to arrhythmias, a causal role in facilitating of the re-entry conduction resulting in ventricular tachycardia/VF causes abnormal cardiomyocyte Cx43 distribution [40]. Accordingly, redistribution of Cx43 from the polar intercalated disc-related position to the lateral sides of cardiomyocytes (in conjunction with reduced Cx43 protein levels and phosphorylation as observed in SHRs) is highly arrhythmogenic and promotes development of VF [19,27,38,41,42]. Interestingly, hypothyroid WKRs that exhibit upregulated Cx43 are much less prone to VF despite the anomalies in Cx43 localization [2,15,17], which was shown in this study as well. It suggests that enhanced Cx43 and its phosphorylation at serine368 are superior to alterations in Cx43 distribution with regard to its propensity to malignant arrhythmias, which is further supported by the results of the most recent study by Kohutova et al. (2019) [37]. Different from WKRs, altered thyroid status is less important in the prediction of lethal arrhythmias in SHRs because there was no effect on total Cx43 protein abundance or its distribution. On the other hand, an increase in P-Cx43 at serine368 in hypothyroid SHRs suggests the need to explore possible antiarrhythmic effects.

Furthermore, our findings also suggest that changes in circulating THs affected ECM markers in WKR but not in SHR hearts. Accordingly, SMAD2/3 protein level, as well as collagen deposition, were increased, and MMP-2 activity was decreased in hypothyroid, whereas the opposite trend was observed in hyperthyroid WKRs. Thus, hypothyroid and hyperthyroid status were characterized by pro-fibrotic and anti-fibrotic phenotypes, respectively. In agreement with this, an upregulation of pro-fibrotic TGF-β1 in hypothyroid Wistar rats has been reported [43]. Fibrosis is caused by excessive deposition of fibrillary collagens due to an imbalance between the collagen production and degradation. Myocardial fibrosis due to hypothyroidism found in this study can be explained by negative regulation of the pro-alfa1 collagen gene expression by THs [44]. However, the absence of fibrosis in hyperthyroidism that was confirmed in this study might be attributed to a decrease in collagen gene expression [20] and enhanced degradative pathways [45,46]. In line with these findings, we have demonstrated the downregulation of TGF-β1 and increase in MMP-2 activity in hyperthyroid WKRs.

Regarding SHRs, it should be noted that the levels of TGF-β1 were lower, SMAD2/3 were similar, and MMP-2 along with collagen deposition were higher than those observed in WKRs; this most likely reflects the adaptation of 12-month-old SHRs to hypertension. Despite significant changes in heart and left ventricle weights indicating myocardial structural remodeling due to altered thyroid status, the ECM markers were unaffected in hearts of SHRs. 

Importantly, our findings suggest that decreased levels total Cx43 and/or P-Cx43 at serine368, although not accompanied by an enhanced interstitial collagen deposition, may promote malignant arrhythmias in hyperthyroid rat heart. Heterogeneously reduced myocardial Cx43 protein abundance has been shown associated with dispersed impulse conduction and electrical instability [47]. In contrast, the heart with upregulated protein level of Cx43 is less prone to arrhythmia despite the presence of collagen accumulation as we have shown in hypothyroid rats. We can conclude that altered thyroid status may affect heart contractile function in both WKRs and SHRs while its effect on the development of malignant arrhythmias is evident in WKRs but hampered in SHRs.

### Limitations of the Study

In the context of our findings showing that responses to altered thyroid status are blunted in SHR versus WKR, it is necessary to acknowledge that the WKR strain does not represent a proper control to the SHR strain, as it was reported by Kurtz and Moritz [48] and Zhang-James et al. [49]. This due to high biological variability of WKR as well as behavioral and genetic differences among the WKR sub-strains obtained from different vendors and breeders. Accordingly, we cannot judge whether findings obtained in the current study would be confirmed when using a more relevant control. Unfortunately, such controls were not commercially available. We used SHR and WKR that differed significantly in blood pressure values, which was one of the key factors when designing of our experiment. Furthermore, both strains exhibited the same values of circulating thyroid hormones of interest in this study. However, it is most likely that WKR do not constitute a single inbred strain and this fact should be taken into consideration regarding our findings. Nevertheless, despite using WKR as a control, our findings warrant further investigation because incidences of thyroid disorders, as well as hypertension, are rising in the human population. Finally, it is unknown whether findings from exact controls would be more relevant to what could be in the human population as phenotypic/genotypic variations in human patients are huge.

## 4. Materials and Methods 

### 4.1. Experimental Design

The experiments were performed using 10-month-old male Wistar Kyoto rats (WKR) and age-matched spontaneously hypertensive rats (SHR) that were housed at 23 ± 1 °C and 12-h light/ dark cycles with ad libitum access to tap water and standard laboratory chow. The maintenance and handling of animals were performed in accordance with the “Guide for the Care and Use of Laboratory Animals” published by US National Institutes of Health (NIH publication No 85-23, revised 1996) and approved by the Ethical Committee of the Institute of Physiology v.v.i., (approval code- 49/17/45.0, approved date- 16 February 2017) Academy of Sciences of the Czech Republic, Prague.

WKR and SHR were randomly divided into three main groups: Euthyroid (EU, *n* = 24), hyperthyroid (HT, *n* = 24) and hypothyroid (HY, *n* = 24), whereby half the animals in each group were treated daily with the preparation of omega-3 polyunsaturated fatty acids (Omacor, containing eicosapentaenoic and docosahexanoic acids, Pronova BioPharma Norge AS, Norway) at a dose of 200 mg/kg b.w. for eight weeks via gavage. Omacor’s 1000 mg capsules contain pure ethyl esters of eicosapentaenoic acid (EPA, 460 mg) plus docosahexaenoic acid (DHA, 380 mg). HT status was induced and maintained until the end of the experiment by intraperitoneal injection of 3,3′,5-triiodo-L-thyronine (T3; Sigma Aldrich, St. Louis, MO, USA) at 0.15 mg/kg b.w. three times weekly and HY status was induced and maintained until the end of the experiment by the addition of 0.05 % solution of methimazole (Sigma Aldrich, St. Louis, MO, USA) to the drinking water. At the end of the experiment rats were overdosed with 100 mg/kg b.w. of anesthetic ketamine (Narketan; Vetoquinol UK Ltd., Towcester, UK) followed by 10 mg/kg b.w. of myorelaxant xylazine (Xylapan; Vetoquinol UK Ltd., Towcester, UK). After blood collection, the heart, liver and thyroid gland were quickly excised; weights of body (BW), thyroid gland (TGW), heart (HW), and left ventricle (LVW) were registered. Myocardial left ventricle tissue (LV) and blood serum samples were stored in a freezer at −80 °C.

### 4.2. Bioactive Parameters Assessed in Blood Samples

Blood glucose levels were obtained using the EasyGluco system (Infopia Co. Ltd., Anyang, South Korea) after overnight fasting at the end of the experiment. Postprandial concentrations of triglycerides, total cholesterol, high-density lipoprotein, and low-density lipoprotein-cholesterol in samples of blood serum were measured using commercial kits (Pliva-Lachema Diagnostika, Brno, Czech Republic). Serum levels of total thyroxine and total triiodothyronine were determined using commercial radioimmunoassay kits (Immunotech/Beckman Coulter Co., Prague, Czech Republic) [50].

### 4.3. Lipid Peroxidation Assay

Level of thiobarbituric acid-reactive substances (TBARS) was determined by reaction with thiobarbituric acid. Briefly, the left heart ventricle homogenates were incubated with trichloroacetic acid and after 10 min of centrifugation, the supernatant was incubated with thiobarbituric and acetic acid at 60 °C for 60 min. After cooling, sample absorbance was measured at 535 nm using a Tecan Infinite M200 plate reader (Männedorf, Switzerland). A calibration curve was prepared using 1,1,3,3-tetraethoxypropane. The results were expressed as nmol of TBARS per mg of protein. Protein levels were measured using the Folin reagent with bovine serum albumin as standard.

### 4.4. Determination of mRNA by Real-Time PCR

RNA was extracted from each LV sample using the Trizol Reagent (Invitrogen, Carlsbad, CA, USA). One microgram of RNA was converted to cDNA using the RevertAidTM H Minus First Strand cDNA Synthesis Kit (Fermentas UAB, Vilnius, Lithuania) with oligo (dT) primers. RealTime PCR was performed on a Light Cycler 480 (Roche Applied Science, Penzberg, Germany) using a PCR reaction mixture containing iQTM SYBR Green Supermix (Bio-Rad, Hercules, CA, USA) plus specific primers according to the manufacturer’s protocol. The level of analyzed transcripts was normalized to the level of the reference gene, hypoxanthine-guanine phosphoribosyltransferase 1 transcript [51]. Specific primers designed by Universal ProbeLibrary (Roche Applied Science) were used, shown in Table 4.

### 4.5. SDS-PAGE and WB analysis of proteins

LV tissue was isolated, weighed, and frozen in liquid nitrogen. Frozen tissue was powdered in liquid nitrogen and subsequently homogenized in SB20 lysis buffer (20% SDS, 10 mmol/L EDTA, 100 mmol/L Tris, pH 6.8). The tissue lysate diluted in Laemmli sample buffer, was boiled for 5 min, and an equal amount of protein was loaded in each well and separated by SDS-PAGE on 10% bis-acrylamide gels at a constant voltage of 120 V (Mini-Protean TetraCell, Bio-Rad). After transferring the proteins to a nitrocellulose membrane (0.2 μm pore size, Advantec, Tokyo, Japan) and blocking for 4 h with 5% fat-free milk in Tris-buffered saline containing 0.1% Tween 20 (TBST), the membrane was incubated overnight with primary antibodies (anti-total-Cx43 (diluted 1:5000, C6219, Sigma-Aldrich); anti-phospho-serine 368-Cx43 (diluted 1:1000, sc-101660, Santa Cruz Biotechnology, Dallas, TX, USA); anti-non-phospho-Cx43 (diluted 1:1000, CX-1B1, ThermoFisher, Waltham, MA, USA); anti-MMP-2 (diluted 1:1000, sc-10736, Santa Cruz); anti-PKC-epsilon (diluted 1:1000, sc-214, Santa Cruz); anti-TGF-β1 (diluted 1:1000, SAB4502954, Sigma-Aldrich); anti-SMAD2/3 (diluted 1:1000, #3102, Cell Signaling Technology, Danvers, MA, USA); and anti-GAPDH (diluted 1:1000, sc-25778, Santa Cruz)). The membrane was subsequently washed in TBST and incubated for 1 h with a horseradish peroxidase-linked secondary antibody (diluted 1:2000, #7074/#7076, Cell Signaling Technology). The electrochemiluminescence method was used for visualization of proteins and the relevant bands were analyzed densitometrically using Carestream Molecular Imaging Software (version 5.0, Carestream Health, New Haven, CT, USA) as described elsewhere [14,37].

### 4.6. Gelatin Zymography 

For determination of MMP-2 activity by zymography, samples from LV tissue and blood serum were homogenized as described previously [52]. Briefly, samples prepared under non-reducing conditions were subjected to SDS-PAGE using gels co-polymerized with gelatin (2 mg/mL). After electrophoresis, gels were washed twice with washing buffer (50 mmol/L Tris-HCl, 2.5% Triton X-100, pH 7.4) and incubated overnight in developing buffer (50 mmol/L Tris-HCl, 10 mmol/L CaCl_2_, 1.25% Triton X-100, pH 7.4) at 37 °C. After incubation, the gels were stained with 1% Coomassie Brilliant Blue G-250 dissolved in an aqueous solution containing 10% acetic acid and 40% methanol and then unstained with an aqueous solution containing 10% acetic acid and 40% methanol. Enzyme activity was visualized as white bands against a dark blue background and analyzed with Carestream Molecular Imaging Software (version 5.0, Carestream Health, New Haven, CT, USA). 

### 4.7. Immunolabeling of Cx43 

As previously described [33,53], LV cryosections (10 μm, cryostat Leica CM1950; Leica Biosystems, Wetzlar, Germany) were washed in phosphate-buffered saline (PBS, 135 mmol/L NaCl, 25 mmol/L KCl, 80 mmol/L Na_2_HPO_4_, 15 mmol/L KH_2_PO_4_; pH 7.4) fixed in ice-cold methanol, permeabilized in 0.3% Triton X-100 in PBS and blocked at room temperature with the solution of 1% bovine serum solution in PBS. Sections were incubated overnight with primary anti-Cx43 antibody (diluted 1:500, MAB 3068, CHEMICON International, Inc., Temecula, CA, USA) and anti-cadherin (diluted 1:300, sc-7939, Santa Cruz) at 4 °C, washed with PBS and subsequently incubated with secondary antibodies conjugated with FITC-fluorescein isothiocyanate (diluted 1:500, Jackson Immuno Research Labs, West Grove, PA, USA) and with Alexa Fluor 594 (diluted 1:500, Jackson Immuno Research Laboratory Labs). Sections washed with PBS were mounted in the Vectashield mounting medium (H-1200, Vector Laboratories-Inc., Burlingame, CA, USA) and examined in Zeiss Apotome 2 microscope (Carl Zeiss, Jena, Germany).

### 4.8. Quantitative Image Analysis of Collagen Deposition

The van Gieson technique was performed to stain collagen deposits. Briefly, heart tissue sections were fixed with 4% buffered formaldehyde followed by washing in PBS and exposure to a mixture of saturated picric acid and 1% aqueous acid fuchsine for 5 min. The red staining of collagen was clearly recognized on the background of the yellow-stained heart muscle. The slices were examined by light microscopy (Axiostar, Carl Zeiss) and acquired images (*n* = 6 per heart) were used for quantitative image analysis using Soft Imaging System GmBH (Münster, Germany). A positive signal was expressed as a proportion of the total tissue area.

### 4.9. Statistical Analysis

Differences between groups were evaluated using one-way analysis of variance (ANOVA) and Bonferroni’s multiple comparison test. The Kolmogorov–Smirnov normality test was used to examine whether variables are normally distributed. Data were expressed as means ± standard deviation (SD); *p* < 0.05 was considered to be statistically significant.

## Figures and Tables

**Figure 1 ijms-20-03758-f001:**
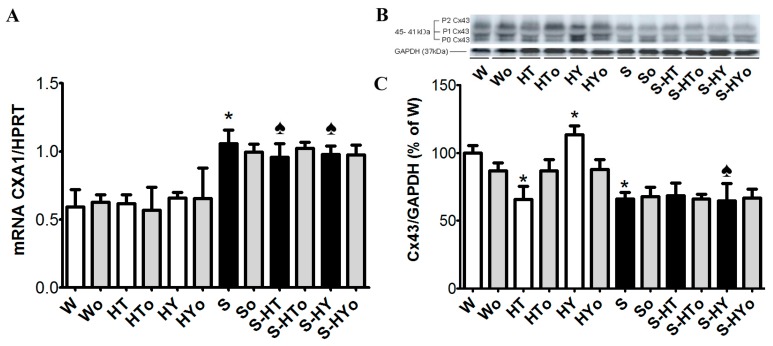
Myocardial expression of *CXA1* mRNA normalized to hypoxanthine guanine phosphoribosyl transferase (*HPRT*) level (**A**), and protein abundance of total myocardial connexin-43 (Cx43) normalized to glyceraldehyde-3-phosphate dehydrogenase (GAPDH) protein level (**B**,**C**) in the male normotensive and hypertensive rats with altered thyroid status with or without feeding with Omacor. W—Wistar Kyoto euthyroid control rats; o—fed with Omacor; HT—hyperthyroid rats; HY—hypothyroid rats; S—spontaneously hypertensive rats; *n* = 6 in each group. Data are presented as means ± SD; * *p* < 0.05 vs. W; ♠ *p* < 0.05 HT vs. S-HT/ HY vs. S-HY.

**Figure 2 ijms-20-03758-f002:**
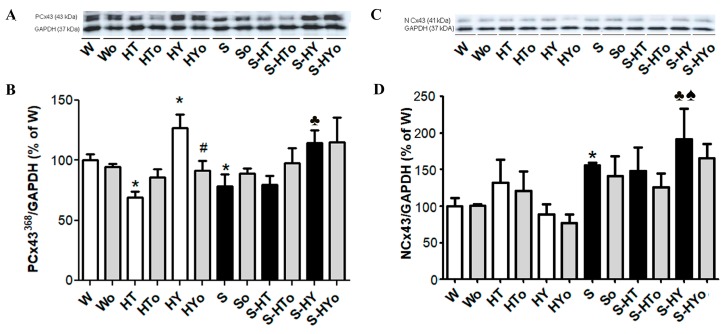
Myocardial protein abundance of the phosphorylated form of connexin-43 (Cx43) specifically on Serine368 (**A**,**B**) and non-phosphorylated form of Cx43 (**C**,**D**) normalized to glyceraldehyde-3-phosphate dehydrogenase (GAPDH) protein level in the male normotensive and hypertensive rats with altered thyroid status with or without feeding with Omacor. W—Wistar Kyoto euthyroid control rats; o—fed with Omacor; HT—hyperthyroid rats; HY—hypothyroid rats; S—spontaneously hypertensive rats; *n* = 6 in each group. Data are presented as means ± SD; * *p* < 0.05 vs. W; ♣ *p* < 0.05 vs. S; # *p* < 0.05 untreated vs. treated with Omacor; ♠ *p* < 0.05 HT vs. S-HT/ HY vs. S-HY.

**Figure 3 ijms-20-03758-f003:**
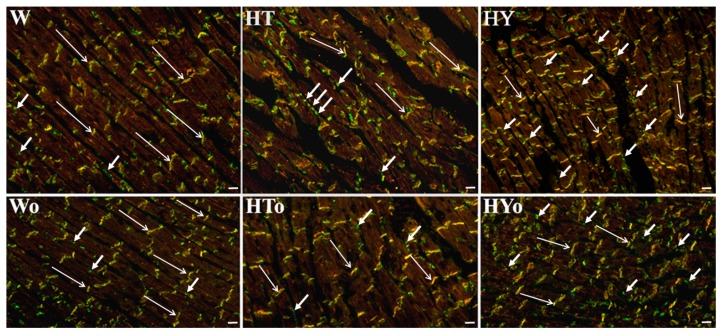
Topology of myocardial connexin-43 (Cx43, green) colocalized with N-cadherin (red) in the male normotensive rats with altered thyroid status with or without feeding with Omacor; W—Wistar Kyoto euthyroid control rats; o—fed with Omacor; HT—hyperthyroid rats; HY—hypothyroid rats. Note conventional immunofluorescence labeling of Cx43 and N-cadherin at the gap junctions (long arrows) and Cx43 on lateral surfaces of the cardiomyocytes (short arrows). Objective 40×, Scale bar 10 µm.

**Figure 4 ijms-20-03758-f004:**
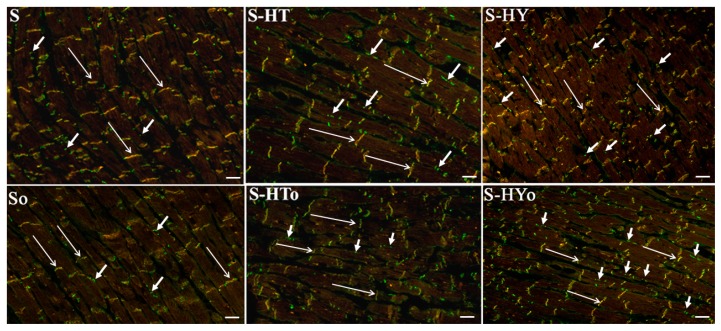
Topology of myocardial connexin-43 (Cx43, green) colocalized with N-cadherin (red) in the male hypertensive rats with altered thyroid status with or without feeding with Omacor; S—spontaneously hypertensive rats; o—fed with Omacor; HT—hyperthyroid rats; HY—hypothyroid rats. Note conventional immunofluorescence labeling of Cx43 and N-cadherin at the gap junctions (long arrows) and Cx43 on lateral surfaces of the cardiomyocytes (short arrows). Objective 40×. Scale bar 10 µm.

**Figure 5 ijms-20-03758-f005:**
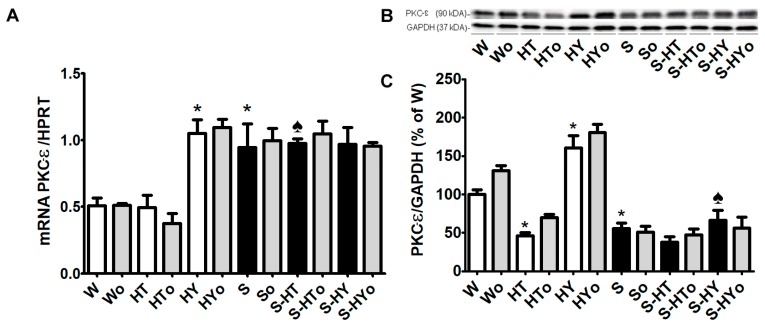
Myocardial expression of protein kinase C epsilon (*PKC-ε*) mRNA normalized to hypoxanthine guanine phosphoribosyl transferase (*HPRT*) level (**A**), and protein abundance of myocardial PKC-ε normalized to glyceraldehyde-3-phosphate dehydrogenase (GAPDH) protein level (**B**,**C**) in the male normotensive and hypertensive rats with altered thyroid status with or without feeding with Omacor. W—Wistar Kyoto euthyroid control rats; o—fed with Omacor; HT—hyperthyroid rats; HY—hypothyroid rats; S—spontaneously hypertensive rats; *n* = 6 in each group. Data are presented as means ± SD; * *p* < 0.05 vs. W; ♠ *p* < 0.05 HT vs. S-HT/ HY vs. S-HY.

**Figure 6 ijms-20-03758-f006:**
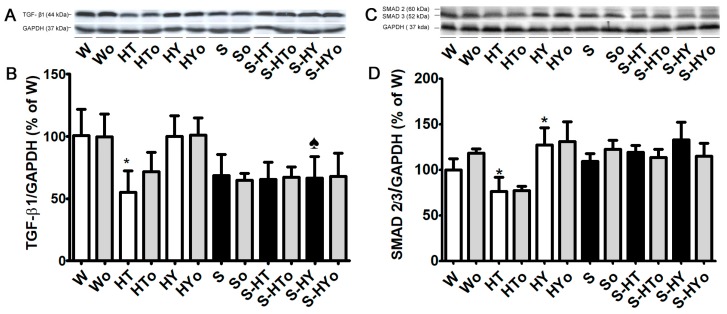
Protein levels of myocardial transforming growth factor β (TGF-β1; **A**,**B**) and TGF transcription factor (SMAD2/3; **C**,**D**) normalized to glyceraldehyde-3-phosphate dehydrogenase (GAPDH) protein level in the male normotensive and hypertensive rats with altered thyroid status with or without feeding with Omacor. W—Wistar Kyoto euthyroid control rats; o—fed with Omacor; HT—hyperthyroid rats; HY—hypothyroid rats; S—spontaneously hypertensive rats; *n* = 6 in each group. Data are presented as means ± SD; * *p* < 0.05 vs. W; ♠ *p* < 0.05 HT vs. S-HT/ HY vs. S-HY.

**Figure 7 ijms-20-03758-f007:**
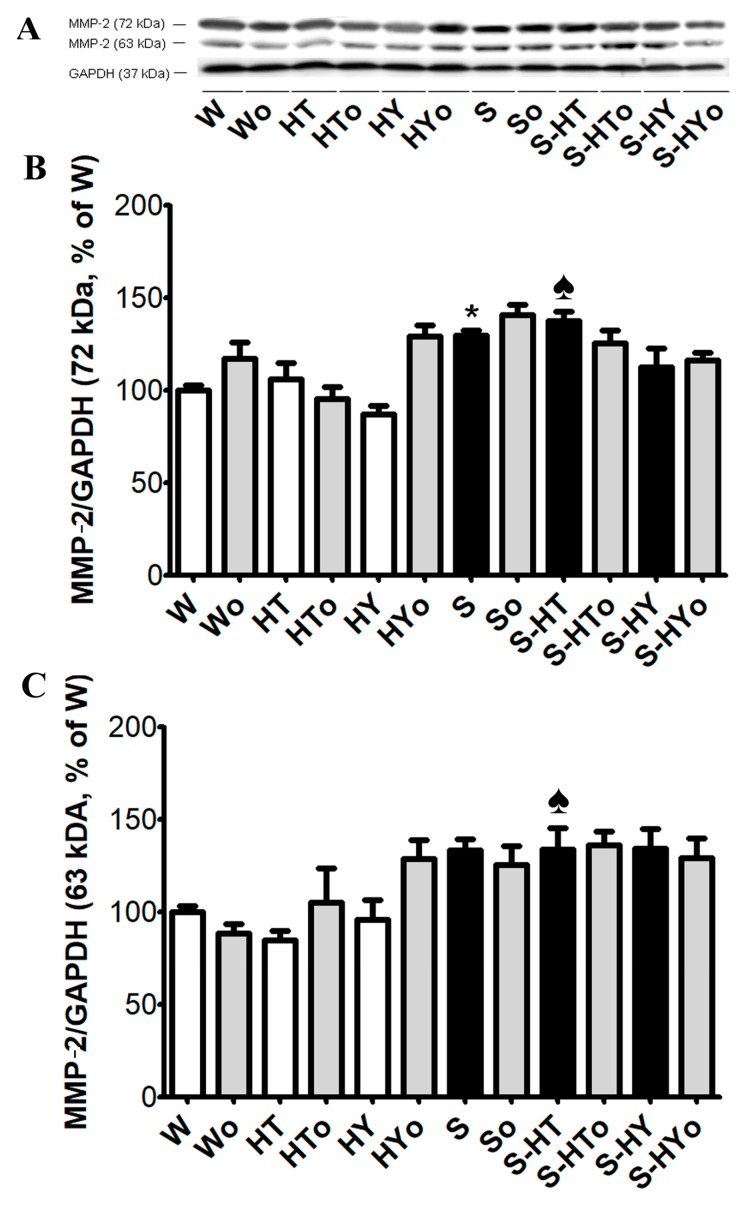
Protein abundance of myocardial matrix metalloproteinase-2 (MMP-2) normalized to glyceraldehyde-3-phosphate dehydrogenase (GAPDH) protein level in the male normotensive and hypertensive rats with altered thyroid status with or without feeding with Omacor. A 72 kDa pro-MMP-2 (**A**,**B**) and 63 kDa active MMP-2 (**A**, **C**) are shown. W—Wistar Kyoto euthyroid control rats; o—fed with Omacor; HT—hyperthyroid rats; HY—hypothyroid rats; S—spontaneously hypertensive rats; *n* = 6 in each group. Data are presented as means ± SD; * *p* < 0.05 vs. W; ♠ *p* < 0.05 HT vs. S-HT/ HY vs. S-HY.

**Figure 8 ijms-20-03758-f008:**
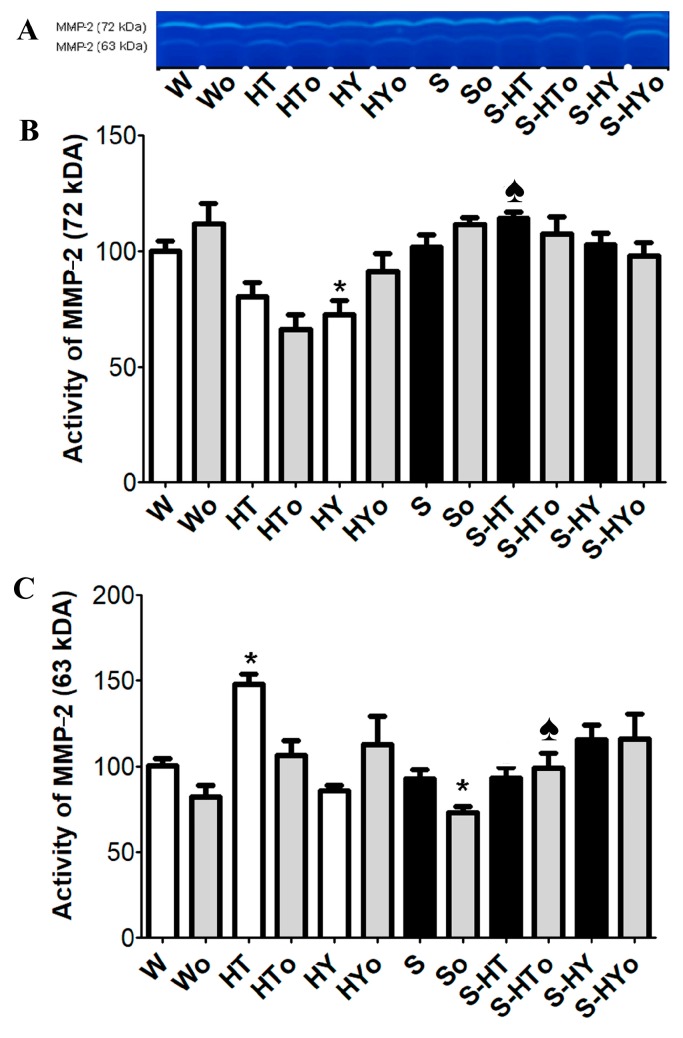
Zymography of matrix metalloproteinase-2 (MMP-2) and quantitative analysis of MMP-2 activity in the male normotensive and hypertensive rats with altered thyroid status with or without feeding with Omacor. 72 kDa pro-MMP-2 (**A**,**B**); 63 kDa active MMP-2 (**A**,**C**); W—Wistar Kyoto euthyroid control rats; o—fed with Omacor; HT—hyperthyroid rats; HY—hypothyroid rats; S—spontaneously hypertensive rats; *n* = 6 in each group. Data are presented as means ± SD; * *p* < 0.05 vs. W; ♠ *p* < 0.05 HT vs. S-HT/ HY vs. S-HY.

**Figure 9 ijms-20-03758-f009:**
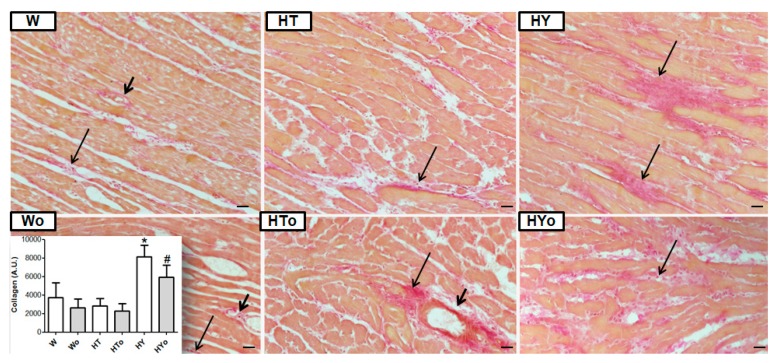
Representative images of van Gieson staining and quantitative analysis of collagen deposition in the male normotensive rats with altered thyroid status, fed with or without Omacor. Interstitial fibrosis—long arrows; Perivascular fibrosis—short arrows. W—Wistar Kyoto euthyroid control rats; o—fed with Omacor; HT—hyperthyroid rats; HY—hypothyroid rats. Objective 40×. Scale bar 10 µm. Data are presented as means ± SD; * *p <* 0.05 vs. W; # *p <* 0.05 untreated vs. treated with Omacor.

**Figure 10 ijms-20-03758-f010:**
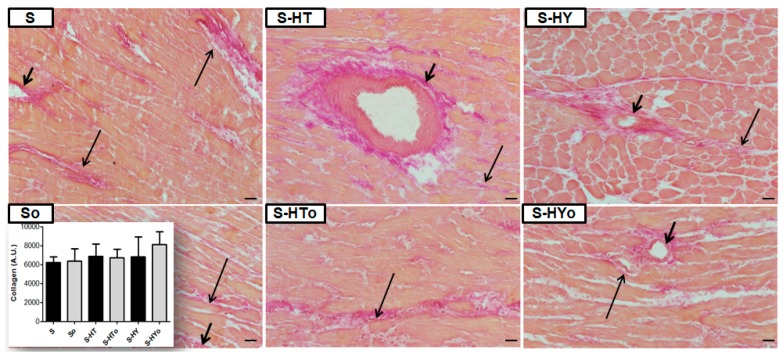
Representative images of van Gieson staining and quantitative analysis of collagen deposition in the male hypertensive rats with altered thyroid status, fed with or without Omacor. Interstitial fibrosis—long arrows; perivascular fibrosis—short arrows. S—spontaneously hypertensive rats; o—fed with Omacor; HT—hyperthyroid rats; HY—hypothyroid rats. Objective 40×. Scale bar 10 µm. Data are presented as means ± SD.

**Table 1 ijms-20-03758-t001:** General characteristics of experimental rats.

Variables	Intake	WKRs	WKR-HTs	WKR-HYs	SHRs	SHR-HTs	SHR-HYs
**T3 (nmol/L)**	**-**	0.88 ± 0.29	1.67 ± 0.55 *	0.26 ± 0.11 *	0.86 ± 0.21	1.61 ± 0.48 ♣	0.21 ± 0.04 ♣
	**Omacor**	1.03 ± 0.26	1.25 ± 0.24	0.20 ± 0.05	0.97 ± 0.23	1.30 ± 0.28	0.16 ± 0.04
**T4 (nmol/L)**	**-**	60.52 ± 7.91	101.71 ± 38.51 *	8.76 ± 4.55 *	39.17 ± 11.29	78.39 ± 20.91 ♣	11.71 ± 2.11
	**Omacor**	54.27 ± 15.81	67.26 ± 10.95 ^#^	5.95 ± 3.32	55.02 ± 10.11	52.63 ± 10.77	6.98 ± 2.65
**BG (mmol/L)**	**-**	4.97 ± 0.59	5.53 ± 0.40	5.97 ± 0.61	6.85 ± 0.58	5.57 ± 0.06	6.50 ± 0.72
	**Omacor**	5.43±1.00	4.80 ± 0.61	5.48 ± 0.59	7.02 ± 0.63	5.85 ± 0.69	5.38 ± 0.58
**TGW (mg)**	**-**	38 ± 3	33 ± 8	94 ± 24 *	22 ± 5	18 ± 4	54 ± 10♣♠
	**Omacor**	51 ± 8	38 ± 10	106 ± 15	31 ± 8	17 ± 6	45 ± 4
**BW (g)**	**-**	444 ± 36	405 ± 27	431 ± 21	410 ± 25	368 ± 22	387 ± 39
	**Omacor**	362 ± 38 ^#^	339 ± 29 ^#^	416 ± 24	433 ± 28	355 ± 10	360 ± 12
**HW (g)**	**-**	1.18 ± 0.08	1.56 ± 0.05 *	0.88 ± 0.12 *	1.72 ± 0.18 *	2.10 ± 0.44 ♣♠	1.24 ± 0.15 ♣♠
	**Omacor**	0.92 ± 0.1	1.49 ± 0.15	0.95 ± 0.06	1.65 ± 0.13	1.88 ± 0.11	1.09 ± 0.15
**LVW (g)**	**-**	0.84 ± 0.05	1.15 ± 0.12 *	0.68 ± 0.05 *	1.32 ± 0.09 *	1.62 ± 0.29 ♣♠	0.98 ± 0.11 ♣♠
	**Omacor**	0.68 ± 0.10	1.02 ± 0.11	0.69 ± 0.06	1.30 ± 0.17	1.51 ± 0.12	0.91 ± 0.09
**TBARSs (nmol/mg)**	**-**	1.42 ± 0.41	2.74 ± 0.45 *	1.43 ± 0.35	1.49 ± 0.15	1.35 ± 0.28	1.31 ± 0.21
	**Omacor**	1.94 ± 0.25	1.94 ± 0.34 ^#^	1.45 ± 0.23	1.83 ± 0.62	1.59 ± 0.23	1.62 ± 0.17

T3—total triiodothyronine; T4—total thyroxine; BG—blood glucose; TGW—thyroid gland weight; BW—body weight; HW—heart weight; LVW—left ventricular weight; TBARSs—thiobarbituric acid reactive substances in myocardium; WKRs—Wistar Kyoto euthyroid control rats; HTs—hyperthyroid rats; HYs—hypothyroid rats; SHRs—spontaneously hypertensive rats; Omacor—omega-3 polyunsaturated fatty acids; *n* = 6 in each group. Data are presented as means ± SD; * *p* < 0.05 vs. WKRs; ♣ *p* < 0.05 vs. SHRs; # *p* < 0.05 untreated vs. treated with Omacor; ♠ *p* < 0.05 HTs vs. SHR-HTs/ HYs vs. SHR-HYs.

**Table 2 ijms-20-03758-t002:** Lipid profile of experimental rats.

Variables	Intake	WKRs	WKR-HTs	WKR-HYs	SHRs	SHR-HTs	SHR-HYs
**TAG (mmol/L)**	**-**	1.18 ± 0.34	0.84 ± 0.14	0.62 ± 0.22 *	0.80 ± 0.19	0.88 ± 0.43	0.52 ± 0.05
	**Omacor**	0.77 ± 0.27	0.53 ± 0.16	0.44 ± 0.05	0.81 ± 0.23	0.55 ± 0.13	1.51 ± 0.04
**HDLchol (mmol/L)**	**-**	1.11 ± 0.13	1.13 ± 0.19	1.66 ± 0.14 *	0.99 ± 0.06	1.07 ± 0.09	1.83 ± 0.41 ♣
	**Omacor**	1.33 ± 0.28	0.83 ± 0.08	1.84 ± 0.26	0.95 ± 0.13	0.87 ± 0.15	2.18 ± 0.12
**LDLchol (mmol/L)**	**-**	0.35 ± 0.08	0.33 ± 0.11	0.80 ± 0.16 *	0.42 ± 0.09	0.52 ± 0.11	1.06 ± 0.15 ♣♠
	**Omacor**	0.54 ± 0.33	0.20 ± 0.05	0.79 ± 0.12	0.44 ± 0.10	0.28 ± 0.10	1.01 ± 0.23
**Tchol/HLDchol**	**-**	1.53 ± 0.11	1.46 ± 0.16	1.56 ± 0.09	1.38 ± 0.57	1.65 ± 0.16	1.65 ± 0.12
	**Omacor**	1.67 ± 0.14	1.38 ± 0.07	1.48 ± 0.05	1.64 ± 0.10	1.44 ± 0.06	1.57 ± 0.09

TAG-triacylglycerol; HDL—high-density lipoprotein; LDL—low-density lipoprotein; Tchol—total cholesterol; WKRs—Wistar Kyoto euthyroid control rats; HTs—hyperthyroid rats; HYs—hypothyroid rats; SHRs—spontaneously hypertensive rats; Omacor—omega-3 polyunsaturated fatty acids; *n* = 6 in each group. Data are presented as means ± SD; * *p* < 0.05 vs. WKRs; ♣ *p* < 0.05 vs. SHRs; # *p* < 0.05 untreated vs. treated with Omacor; ♠ *p* < 0.05 HTs vs. SHR-HTs/HYs vs. SHR-HYs.

**Table 3 ijms-20-03758-t003:** Myocardial mRNA transcripts level in the male experimental rats.

Variables	Intake	WKRs	WKR-HTs	WKR-HYs	SHRs	SHR-HTs	SHR-HYs
**mRNA *MYH7***	**-**	37.78 ± 4.39	3.55 ± 0.95 *	132.60 ± 41.59 *	237.40 ± 67.27 *	48.161 ± 25.781 ♣	368.82 ± 97.55 ♣♠
	**Omacor**	22.94 ± 3.95	4.64 ± 2.15	81.24 ± 6.78	246.10 ± 91.50	68.462 ± 22.53	441.72 ± 39.92
**mRNA *SERCA2***	**-**	43.77 ± 7.60	61.52 ± 16.26	20.08 ± 3.95 *	36.07 ± 2.36	45.07 ± 5.09	13.42 ± 5.09
	**Omacor**	40.95 ± 9.99	46.71 ± 12.72	19.22 ± 5.04	31.20 ± 13.47	32.47 ± 5.94	15.35 ± 4.44
**mRNA *PPARγ***	**-**	0.03 ± 0.01	0.06 ± 0.02 *	0.02 ± 0.01	0.03 ± 0.01	0.05 ± 0.002	0.02 ± 0.002
	**Omacor**	0.05 ± 0.01	0.05 ± 0.02	0.03 ± 0.01	0.02 ± 0.01	0.04 ± 0.004	0.02 ± 0.003

*MYH7*—myosin heavy chain β; *SERCA2*—sarcoplasmic Ca^2+^ ATPase; *PPARγ*—peroxisome proliferator-activated receptors gamma; WKRs—Wistar Kyoto euthyroid control rats; HTs—hyperthyroid rats; HYs—hypothyroid rats; SHRs—spontaneously hypertensive rats; Omacor—omega-3 polyunsaturated fatty acids; *n* = 6 in each group. The values of mRNA were normalized to the reference gene hypoxanthine guanine phosphoribosyl transferase (*HPRT*). Data are presented as means ± SD; * *p* < 0.05 vs. WKRs; ♣ *p* < 0.05 vs. SHRs; ♠ *p* < 0.05 HTs vs. SHR-HTs/ HYs vs. SHR-HYs.

**Table 4 ijms-20-03758-t004:** Primer sequences for messenger RNA transcript analysis.

Transcript	Forward (5′→3′)	Reverse (5′→3′)
Connexin-43	AGCCTGAACTCTCATTTTTCCTT	CCATGTCTGGGCACCTCT
Protein kinase C epsilon	AAACACCCTTATCTAACCCAACTCT	CATATTCCATGACGAAGAAGAGC
β-myosin heavy chain	GCACCGTGGACTACAATATCCT	5CTTAGGAGCTTGAGGGAGGAC
Sarcoplasmic reticulum Ca2+ATPase	ACCTGGAAGATTCTGCGAAC	AATCCTGGGAGGGTCCAG
Peroxisome proliferator-activated receptor-γ	CCCAATGGTTGCTGATTACA	GGACGCAGGCTCTACTTTGA
Hypoxanthine guanine phosphoribosyl transferase	GACCGGTTCTGTCATGTCG	ACCTGGTTCATCATCACTAATCAC

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
