# Peer review of "Cardiac Cx43 and ECM Responses to Altered Thyroid Status Are Blunted in Spontaneously Hypertensive versus Normotensive Rats"

_ijms, 2019, doi:10.3390/ijms20153758_

Round 1

Reviewer 1 Report

Sykora et al. investigate the response of hypertensive SHR rats to a hypo- or hyperthyroid state in comparison to non-hypertensive control rats (Kyoto Wister). They found that while control rats display the expected reaction such as up or downregulation of Cx43 protein and Cx43 phosphorylation by PKC-εas well as ECM responses, such reactions were blunted in the SHR rats.

The study is well written and sound. Experiments were performed carefully and statistics appear correct.

 However, I have two major points that need to be considered:

1.)  The described phenotype is strong and a definitely interesting observation. However, the manuscript is very descriptive and I do not understand how hypertension is mechanistically linked to the phenotype. The authors should investigate this point in more detail. The impact of TH on Cx43 phosphorylation via PKC has been previously described by the authors (Lin, 2008, Bacova, 2016, Mitasikova, 2009, Bacova, 2017). For that reason, besides the observation that hypertension blunts this reaction, the study does not present many new information.

2.) I have serious doubts with the general design of the study. Using Kyoto Wistar rats as a control for SHR rats has been questioned since the 1990`s. Although both lines are derived from the same genetic background, they split in the 1960`s and are maintained since then as separate colonies. Several publications suggest that using Kyoto Wistar rats as control for SHR appear problematic as they differ from each other in several aspects (as also found by the authors -> see table 1 and 2) and some colonies of these animals were even described to be genetically different (H`Doubler et al., 1991; Zhang-James et al., 2013). Of note, great variations in between different colonies of Kyoto Wister rats were found depending on the supplier of these animals (Kurtz et al., 1987). This could potentially have serious effects on the study. A good example are the C57Bl/6N vs. C57Bl6/J mouse strains. Both are considered to be wildtype. However, although derived from the same genetic background, C57Bl/6J shows a deficiency in the Nnt-gene (coding nicotinamide nucleotide transhydrogenase) and is protected from oxidative stress and therefore heart failure (Nickel et al., 2015). 

It is very important to consider, that mouse or rat strains, although originally coming from the same genetic background, show a high variability and might be hard to compare (https://www.jax.org/news-and-insights/jax-blog/2016/june/there-is-no-such-thing-as-a-b6-mouse). This is of high importance, especially in this study, where the exact mechanism explaining the phenotype is unclear.

The authors must consider this point and critically discuss it in the context of their study.

Author Response

Reviewer 1.

 Open Review

English language and style

( ) Extensive editing of English language and style required
( ) Moderate English changes required
(x) English language and style are fine/minor spell check required
( ) I don't feel qualified to judge about the English language and style

Yes

Can be improved

Must be improved

Not applicable

Does the introduction provide sufficient background and include all   relevant references?

(x)

( )

( )

( )

Is the research design appropriate?

( )

( )

( )

( )

Are the methods adequately described?

( )

( )

( )

( )

Are the results clearly presented?

( )

( )

( )

( )

Are the conclusions supported by the results?

( )

( )

( )

( )

Comments and Suggestions for Authors

Sykora et al. investigate the response of hypertensive SHR rats to a hypo- or hyperthyroid state in comparison to non-hypertensive control rats (Kyoto Wister). They found that while control rats display the expected reaction such as up or downregulation of Cx43 protein and Cx43 phosphorylation by PKC-εas well as ECM responses, such reactions were blunted in the SHR rats.

The study is well written and sound. Experiments were performed carefully and statistics appear correct.

However, I have two major points that need to be considered:

1.)    The described phenotype is strong and a definitely interesting observation. However, the manuscript is very descriptive and I do not understand how hypertension is mechanistically linked to the phenotype. The authors should investigate this point in more detail. The impact of TH on Cx43 phosphorylation via PKC has been previously described by the authors (Lin, 2008, Bacova, 2016, Mitasikova, 2009, Bacova, 2017). For that reason, besides the observation that hypertension blunts this reaction, the study does not present many new information.

First of all, thank you very much for your time to review our manuscript. We indeed appreciate your highly professional approach. We agree with you that our study is rather descriptive. We have noted in the first version of the manuscript in Discussion that we should investigate more in details mechanisms that can explain why the cardiac responses of hypertensive rats to altered thyroid status are blunted comparing to non-hypertensive animals. Questions arise as whether genomic or nongenomic (or both) signaling of thyroid hormones is blunted in SHR. Cx43 transcript was not affected due to altered thyroid status regardless the strains. However, Cx43 protein levels were changed in hypo- and hyper-thyroid WKY while not in SHR. Likewise functional phosphorylated forms of Cx43 were not suppressed in hyperthyroid SHR unlike WKY.  These facts suggest that responsiveness of SHR heart to altered thyroid status is hampered post-translationally. To go deeply it is necessary to include wider spectrum of analysis, namely phosphorylation of Cx43 by various protein kinases. Phosphorylation of Cx43 is known to affect profoundly Cx43 turnover, which is fast (half-life 1.5 hr) in the heart (Dhein et al. 1998, Saffitz et al. 2000). Another factor implicated in hampered response of SHR to alterations of TH could be the difference in composition of fatty acids, namely omega-3 PUFA, in cell membranes as indicated by lower omega-3 index in SHR versus Wistar rats (Bacova et al. 2013, Tribulova et al. 2017).

Goal of our study was first of all to examine responsiveness of SHR heart to altered thyroid status focusing on targeted myocardial proteins implicated in development of malignant arrhythmias. While taking into consideration two facts: 1/ SHRs are prone to such arrhythmias due to Cx43 and ECM abnormalities (as shown in our previous studies as well in review Egan Benova et al. 2016; 2/essential hypertension in humans increases a risk for lethal arrhythmias whereby Cx43 and ECM abnormalities might be implicated (Egan Benova et al. 2016) . We hope that our findings of blunted response of SHR to hyperthyroid and hypothyroid status can challenge further research. Just as you suggested this research should be focused more mechanistically along with using appropriate controls.  To be open, we are “quite happy” that altered thyroid status did not enhance deterioration of SHR heart.

2.) I have serious doubts with the general design of the study. Using Kyoto Wistar rats as a control for SHR rats has been questioned since the 1990`s. Although both lines are derived from the same genetic background, they split in the 1960`s and are maintained since then as separate colonies. Several publications suggest that using Kyoto Wistar rats as control for SHR appear problematic as they differ from each other in several aspects (as also found by the authors -> see table 1 and 2) and some colonies of these animals were even described to be genetically different (H`Doubler et al., 1991; Zhang-James et al., 2013). Of note, great variations in between different colonies of Kyoto Wister rats were found depending on the supplier of these animals (Kurtz et al., 1987). This could potentially have serious effects on the study. A good example are the C57Bl/6N vs. C57Bl6/J mouse strains. Both are considered to be wildtype. However, although derived from the same genetic background, C57Bl/6J shows a deficiency in the Nnt-gene (coding nicotinamide nucleotide transhydrogenase) and is protected from oxidative stress and therefore heart failure (Nickel et al., 2015). It is very important to consider, that mouse or rat strains, although originally coming from the same genetic background, show a high variability and might be hard to compare (https://www.jax.org/news-and-insights/jax-blog/2016/june/there-is-no-such-thing-as-a-b6-mouse). This is of high importance, especially in this study, where the exact mechanism explaining the phenotype is unclear. The authors must consider this point and critically discuss it in the context of their study.

We are very thankful for your attention and important comments dealing with biological variability of WKY rats (Kurtz and Morris 1987) as well as behavioral and genetic differences among the WKY sub-strains from the different vendors and breeders (Zhang-James et al. 2013).

Accordingly, we included these relevant notes in the paragraph Limitations of the study. We agree fully with you that any study using SHR should interpret obtained findings taking into consideration above mentioned facts.  Currently, most researchers still would have a problem to use appropriate controls to SHR strain due to poor availability. According to Kurtz and Morris the best approach probably is to use as a control the WKY rat strain, the base normotensive stock from which the SHR rat was derived. However, is it such approach real? Or is it possible to select and to buy sub-strains as suggested by Zhang-James et al.? With technical advances in microarray genotyping and genome sequencing, we can now use whole genome information to sort out the genetic relationships of different rat sub-strains, including the WKY and SHR. Nevertheless, it seems that it is still limited approach.

We used SHR and WKY which differ significantly in the values of BP. It was one of the key factors when making design of our experiment. Furthermore, WKY and SHR, exhibit the same values of circulating thyroid hormones that we took into consideration. However, it is most likely that rats designated WKY do not constitute a single inbred strain. This fact we took into consideration when interpreting our findings. We included these notes in the paragraph of Limitation

We think that despite using WKY as a controls our findings may challenge further investigation because disorders of thyroid status as well as hypertension are rising in human population. We also would like to note that we are not sure as whether findings from exact controls would be more relevant to what we can expect in human population. Phenotypic/genotypic variations in humans and patients are huge. What about, representative controls in clinical trials? According to our experience, we are convinced that numerous factors can play a role in responsiveness to stressful conditions. Examined compound or intervention may exert opposite effects even in the same strain. There is no question what is the true but what is behind? Finally, we would like to note the fact that the responses to altered thyroid status were similar in WKY and Wistar rats (as previously published) despite some sub-strain differences.

We hope that revised manuscript fits with your remarks and thoughts, otherwise we are still ready to work on it.

Unfortunately, we have used our samples that we shared with coauthors and we are not able to perform additional analysis.

Reviewer 2 Report

The Authors addressed an interesting issue of cardiac connexin-43 and extracellular matrix responses to thyroid hormones in spontaneously hypertensive and normotensive rats. However, certain issues should be addressed more carefully to fully appreciate the work.

1. In the Introduction it is not clear why the Authors refer to MYH7 and SERCA2 (lines 68-71) and what was the basis for the hypothesis formulated in the lines 72 – 73. This should be rewritten and rephrased to clearly and logically explain how the Authors reached the hypothesis that the hypertensive rats differ in responsiveness to thyroid hormones as compared to normotensive rats. What does “particularly diseased heart” mean (line 68)?

2. Throughout the text the Authors incorrectly use the term “protein expression” whereas expression should be used for transcription processes (genes, mRNA); the proper description of translation requires words such as “protein synthesis”, “protein abundance”, etc. Proper use of the expression/synthesis permits the reader not the guess what is the meaning of “The expression of this protein kinase was significantly lower, whereas that of its transcript was higher in SHRs than in WKRs” (lines 180-182). Why the Authors do not discuss this result in Discussion? Also, names of genes should be written in italics.

3. In the sentence opening 2.7. chapter apparently there is something missing: “As shown in Fig. 9, the red staining attributed to collagen deposition scarcely present in hyperthyroid but enhanced in hypothyroid WKR heart LV.” Also, in the next sentence (line 235) the phrase “the collagen was significantly enhanced” is awkward; maybe “collagen content” or “collagen amount” would sound better?

4. The Discussion is very difficult to follow, which in part is caused by not a clear indication of the Authors data and the previously published information. Can the presented data be used to better understand the effects of thyroid dysfunction on the cardiovascular system in humans?

5. The manuscript should be carefully edited for English.

Author Response

Reviewer 2

Yes

Can be improved

Must be improved

Not applicable

Does the introduction provide sufficient background and include all   relevant references?

( )

( )

(x)

( )

Is the research design appropriate?

(x)

( )

( )

( )

Are the methods adequately described?

(x)

( )

( )

( )

Are the results clearly presented?

( )

( )

(x)

( )

Are the conclusions supported by the results?

( )

( )

(x)

( )

Comments and Suggestions for Authors

The Authors addressed an interesting issue of cardiac connexin-43 and extracellular matrix responses to thyroid hormones in spontaneously hypertensive and normotensive rats. However, certain issues should be addressed more carefully to fully appreciate the work.

Thank you very much for your time to review our manuscript as well as for your relevant comments and suggestions. We did our best to improve our manuscript following your stimuli.

1. In the Introduction it is not clear why the Authors refer to MYH7 and SERCA2 (lines 68-71) and what was the basis for the hypothesis formulated in the lines 72 – 73. This should be rewritten and rephrased to clearly and logically explain how the Authors reached the hypothesis that the hypertensive rats differ in responsiveness to thyroid hormones as compared to normotensive rats. What does “particularly diseased heart” mean (line 68)?

We did revision of the third paragraph to make it more clear that the basis for our hypothesis were the differences in myocardial abnormalities induced by hypertension versus altered thyroid status. We displaced “diseased” heart for “in the heart during pathophysiological condition”. 

2. Throughout the text the Authors incorrectly use the term “protein expression” whereas expression should be used for transcription processes (genes, mRNA); the proper description of translation requires words such as “protein synthesis”, “protein abundance”, etc. Proper use of the expression/synthesis permits the reader not the guess what is the meaning of “The expression of this protein kinase was significantly lower, whereas that of its transcript was higher in SHRs than in WKRs” (lines 180-182). Why the Authors do not discuss this result in Discussion? Also, names of genes should be written in italics.

Thanks to your important comment we used proper term for changes in proteins levels throughout the text.  We included in discussion that although protein kinase transcript was higher in SHR versus WKY the abundance of protein was lower most likely due to some disorder in translation and/or turnover (higher degradation).  We did use italic for genes.

3. In the sentence opening 2.7. chapter apparently there is something missing: “As shown in Fig. 9, the red staining attributed to collagen deposition scarcely present in hyperthyroid but enhanced in hypothyroid WKR heart LV.” Also, in the next sentence (line 235) the phrase “the collagen was significantly enhanced” is awkward; maybe “collagen content” or “collagen amount” would sound better?

Thanks for showing us mistakes despite all authors have read manuscript. We did make corrections.

4. The Discussion is very difficult to follow, which in part is caused by not a clear indication of the Authors data and the previously published information. Can the presented data be used to better understand the effects of thyroid dysfunction on the cardiovascular system in humans?

We tried to revise Discussion to emphasize novel findings of the current study. We also included those findings, which were consistent with previously published data. We hope that revised version is more clear. We think that findings obtained in this study may encourage physicians to perform examinations as whether hypothyroid status promote cardiac fibrosis (using gadolinium-enhanced magnetic resonance imaging) and hyperthyroid status cardiac arrhythmias (using electrophysiological protocol) in their patients.

Myocardial late gadolinium enhancement on contrast-enhanced magnetic resonance imaging of patients with hypertrophic cardiomyopathy has been suggested to represent intramyocardial fibrosis and, as such, an adverse prognostic risk factor (Rubinshtein et al 2010) .

Electrophysiological testing is routinely performed in conscious humans to evaluate arrhythmias and disorders of cardiac conduction (Lujan and diCarlo 2014).

5. The manuscript should be carefully edited for English.

We again underwent manuscript for English editing hoping for improvement.

Round 2

Reviewer 1 Report

no further comments

Reviewer 2 Report

Thank you for the corrections and explanations. They are satisfactory.